Exploring the socio-economic and environmental components of infectious diseases using multivariate geovisualization: West Nile Virus

Kala Abhishek K. 1 2
Atkinson Samuel F. 1 2 atkinson@unt.edu
Tiwari Chetan 1 3
1 Advanced Environmental Research Institute, University of North Texas , Denton, TX , USA
2 Department of Biological Sciences, University of North Texas , Denton, TX , USA
3 Department of Geography and the Environment, University of North Texas , Denton, TX , USA
Wang Lei
Electronic publication date: 2020 Jul 27
Publication date: 2020
Volume: 8
Electronic Location ID: e9577
Received 2020 Mar 2; Accepted 2020 Jun 29
Copyright: © 2020 Kala et al.
Copyright year: 2020
Copyright holder: Kala et al.
License: This is an open access article distributed under the terms of the Creative Commons Attribution License, which permits unrestricted use, distribution, reproduction and adaptation in any medium and for any purpose provided that it is properly attributed. For attribution, the original author(s), title, publication source (PeerJ) and either DOI or URL of the article must be cited.
License URL: https://creativecommons.org/licenses/by/4.0/

Keywords: West Nile Virus, Public health, Self organizing maps, Parallel coordinate plots, Data mining

Funding: University of North Texas This work was supported in part by the Advanced Environmental Research Institute, the Department of Biological Sciences, and the Department of Geography and the Environment, all of the University of North Texas. The funders had no role in study design, data collection and analysis, decision to publish, or preparation of the manuscript.

==============================
Background

This study postulates that underlying environmental conditions and a susceptible population’s socio-economic status should be explored simultaneously to adequately understand a vector borne disease infection risk. Here we focus on West Nile Virus (WNV), a mosquito borne pathogen, as a case study for spatial data visualization of environmental characteristics of a vector’s habitat alongside human demographic composition for understanding potential public health risks of infectious disease. Multiple efforts have attempted to predict WNV environmental risk, while others have documented factors related to human vulnerability to the disease. However, analytical modeling that combines the two is difficult due to the number of potential explanatory variables, varying spatial resolutions of available data, and differing research questions that drove the initial data collection. We propose that the use of geovisualization may provide a glimpse into the large number of potential variables influencing the disease and help distill them into a smaller number that might reveal hidden and unknown patterns. This geovisual look at the data might then guide development of analytical models that can combine environmental and socio-economic data.

Methods

Geovisualization was used to integrate an environmental model of the disease vector’s habitat alongside human risk factors derived from socio-economic variables. County level WNV incidence rates from California, USA, were used to define a geographically constrained study area where environmental and socio-economic data were extracted from 1,133 census tracts. A previously developed mosquito habitat model that was significantly related to WNV infected dead birds was used to describe the environmental components of the study area. Self-organizing maps found 49 clusters, each of which contained census tracts that were more similar to each other in terms of WNV environmental and socio-economic data. Parallel coordinate plots permitted visualization of each cluster’s data, uncovering patterns that allowed final census tract mapping exposing complex spatial patterns contained within the clusters.

Results

Our results suggest that simultaneously visualizing environmental and socio-economic data supports a fuller understanding of the underlying spatial processes for risks to vector-borne disease. Unexpected patterns were revealed in our study that would be useful for developing future multilevel analytical models. For example, when the cluster that contained census tracts with the highest median age was examined, it was determined that those census tracts only contained moderate mosquito habitat risk. Likewise, the cluster that contained census tracts with the highest mosquito habitat risk had populations with moderate median age. Finally, the cluster that contained census tracts with the highest WNV human incidence rates had unexpectedly low mosquito habitat risk.

Introduction

Variations in infectious disease risk occur across environmental gradients and population groups. Such variations often manifest themselves in geographic space and can be attributed to complex interactions between the environment, population, and behavior (Meade, 1977). The underlying processes behind these interactions occur at different, and often conflicting spatial and temporal scales. Additionally, the data related to those processes are often collected within different spatial boundaries (e.g., county level versus census tract verses ecosystem) and for differing research purposes. Attempts to understand these processes by exploring primary or secondary data sources can introduce additional levels of complexity due to issues of uncertain data collection contexts (Kwan, 2012), incomplete or unavailable data (Zhang & Goodchild, 2002), or differences in the underlying questions that drove data collection in the first place (Elliott & Wartenberg, 2004). Further, traditional approaches to model any single complex process of disease risk, including confirmatory or validation steps, rely on well-defined outcome measures and a set of clearly specified dependent and independent variables. In order to explore questions that require coupling more than one complex process simultaneously, traditional modeling techniques do not appear to be directly applicable and will likely need modifications to be useful.

Diez Roux & Mair (2010) show that risk can be expressed at different spatial scales and argue that differential disease risks occur across individual- and group-level characteristics. Individual characteristics include attributes of the individuals at risk (e.g., age, gender, income and other personal attributes) while group-level characteristics include the environmental and socio-economic/demographic context of places to which those populations belong (e.g., vector habitat conditions, socio-economic and demographic profiles, and climatic conditions). Collecting data on these characteristics are driven by various primary questions, measured at various scales, and reported for various purposes. While the disease ecology triangle (Meade, 1977) provides a robust framework for studying interactions between human populations, disease agents, and the environment, it is important to recognize that analytical studies of public health risk must find mechanisms to reconcile process, scale, and data complexity. Some researchers approach this problem with complex multilevel models (multiple scales, misalignment of spatial and temporal boundaries, uncertain research context of the data—see Diez-Roux (2000), (Edsall, MacEachren & Pickle, 2001; Edsall, 2003a, 2003b; Kraak & Madzudzo, 2007). However, in this article, we suggest that prior to developing these analytical models to combine multiple complex processes, exploratory approaches that use geovisualization techniques may provide valuable insights for identifying variables and associated processes that contribute to variations in disease risk across space and time. These types of insights may prove to be a valuable intermediate step between models that explore environmental determinants of infectious disease risk, or models that explore demographic determinants of infectious disease vulnerability, and more complex models that combine both environmental and demographic variables. We build upon our previous study that used a spatially-explicit environmental model to assess West Nile Virus (WNV) risk in California based on the relationship between WNV incidence and mosquito habitat suitability, and here we report on the visualization of “population” or “human” components of the disease ecology framework (Meade, 1977) with a spatial lens. While the environmental data used in our earlier model were of relatively fine geographic resolution (all layers resampled to 120 m cell size), the study was limited to a county-level analysis due to the non-availability of both fine-scale WNV disease human incidence data and a related, surrogate parameter, WNV infected dead bird data reported at the county level. As many public health researchers lament, our previous analysis would have been more valuable if WNV human incidence data were available at a finer spatial resolution such as census block groups or tract (DeGroote et al., 2008). None-the-less, we suggest that geovisualization techniques can be used to overcome some of these data limitations by enabling hypothesis generation, seeding confirmatory modeling approaches, and aiding public health practice by providing a platform for exploring complex interactions between the disease, the environment within which it operates, and the populations impacted.

West Nile Virus, a vector-borne disease that is primarily spread by the Culex species of mosquitos, was first detected in the United States in 1999 (Nash et al., 2001). Several studies have utilized the information from satellite imagery for environmental characteristics such as temperature, vegetation cover, and moisture (Ozdenerol, Bialkowska-Jelinska & Taff, 2008; Rodgers & Mather, 2014). Land surface temperature was attributed as one of the main factors contributing to the WNV propagation in Southern California (Liu & Weng, 2012). They associated higher temperature to viral replication in mosquitoes and related lower elevations as more susceptible to WNV invasion due to warmer temperatures in coastal plains habitats (Wimberly et al., 2008). Mean temperature during summers, land surface temperature, elevation, diversity of landscape, and water content in vegetation were the main environmental factors contributing to WNV propagation in Southern California. High temperature has been consistently associated with outbreaks and hotspots of WNV activity (Hartley et al., 2012; Reisen, Fang & Martinez, 2014; Hoover & Barker, 2016), some studies have suggested that certain mosquito species are associated with more urban habitats (Reisen et al., 2008; Kilpatrick, 2011; Savage et al., 2014), some have linked drought to WNV outbreaks (Paz, 2015; Paull et al., 2017), and others (e.g., Cooke, Grala & Wallis, 2006) have explored the connections between WNV human infection risk and environmental conditions such as the presence of streams, vegetation, and roads.

In our earlier study (Kala et al., 2017) geostatistical and spatial analysis techniques were used to build a spatially explicit model after exploring multiple environmental factors (i.e., factors directly or indirectly related to known mosquito determinants such as vegetation, elevation, evapotranspiration, streams, land use and temperature) that linked mosquito habitat suitability to the number of WNV-positive dead birds, which was used as a surrogate for human WNV risk. That study concluded that including spatial heterogeneity in the modeling improved predictive ability in understanding WNV risk. A geographically weighted regression (GWR) was applied to a statistically significant ordinary least squares (OLS) model to improve model fit from 61% to 71%. The resulting WNV disease risk surface was created using multi-criteria decision analysis approach (detailed process can be referred to our previous article, Kala et al. (2017)). This modeling process was based upon four steps: (1) establishment of the environmental factors, (2) standardization of the factors, (3) establishment of relative weights for each factor, and (4) a Simple Additive Weighting (SAW) method to construct the disease risk surface.

Ruiz et al. (2004) reported that socio-economic factors such as age, income, and race/ethnicity of the human population can also be important predictors of WNV infection risk in humans. While many attempts to predict the risk of WNV transmission have been published, efforts that attempt to link both environmental and socio-economic factors within a spatial framework have resulted in less than complete understanding of the complex relationships associated with human infection risk of WNV. In the study reported here we hypothesize that geovisualization techniques to explore the relationships between disease outcomes, population characteristics, and the environment within which they interact will result in a more complete understanding of the complex patterns related to this disease. A more complete understanding may open doors to more traditional model development and validation approaches that are familiar to public health planners.

This hypothesis suggests that an integrated approach to understanding the relationships of environmental variables and human population demographics on WNV risk should improve our ability to explore large numbers of possible combinations of the processes in order to discover potential hidden but useful patterns. However, Guo et al. (2005) asserted that even in a selected subset of the data it is still a challenge to discover hidden relationships as potential patterns may be expressed in various forms—perhaps linear or non-linear, perhaps spatial or non-spatial, or perhaps some such combination. Geovisualization tools can be useful to support multivariate analysis of geospatial data in order to highlight these potential patterns. We have attempted to add value to our earlier GWR model by including information on the spatial characteristics of human population via geovisualization. The addition of demographic data alongside the environmental model may provide understanding to public health planners who want to better understand patterns related to an infectious disease. This added value was accomplished with geovisualization tools to develop self-organizing maps (SOM) and parallel coordinate plots (PCP) to provide insights into the complex processes that operate simultaneously across environmental and socio-economic patterns of this public health issue.

Materials and Methods

Disease vectors and pathogen reservoirs typically intersect within the context of specific environmental factors (Rochlin et al., 2011), while the risk of host infection is influenced by the composition of a susceptible population. For the mosquito vector, the WNV pathogen and the human host population, environmental and socio-economic factors that have been identified by previous research studies were utilized in this study. Several studies have utilized mosquito habitat suitability as a surrogate for estimating WNV risk for human infection (e.g., Cooke, Grala & Wallis, 2006). In the study reported here, our earlier mosquito habitat suitability model was used to describe the environmental processes occurring in our study area, while census tract level demographic data were used to describe the socio-economic processes at play (see Table 1). Figure 1 illustrates the model framework including the advantages of using this approach.

Table 1 Variables related to susceptible human population characteristics (composition) and vector habitat characteristics (context) utilized in this study.

Human population characteristics (demographic composition)	Mosquito habitat characteristics (environmental context)	
Factors studied
(reference)	Relation to WNV risk	Factors studied
(reference)	Relation to WNV risk	
Old age
(Jean et al., 2007; Ruiz et al., 2004)	Weakened immune system	Stream, Vegetation, Road
(Cooke, Grala & Wallis, 2006; Kala et al., 2017)	Sites for breeding and resting.	
Male sex
(Murray et al., 2006)	Social history or lifestyle.	Temperature
(Kala et al., 2017; Wimberly et al., 2008)	Increases growth rate of vector, decreases egg development cycle and shortens extrinsic incubation period of vector.	
Race/Ethnicity
(Ruiz et al., 2004)	Increased risk from behaviors linked to their lifestyle.	Surface slope
(Ozdenerol, Bialkowska-Jelinska & Taff, 2008)	Water stagnation creating mosquito breeding ground.	
Income
(Ruiz et al., 2004)	Increased risk from behaviors linked to their lifestyle.	Cultivated land, Developed land
(Kilpatrick, 2011)	Preferred natural ground pools in cultivated land and warmer micro-climates in developed lands.	

Figure 1 West Nile Virus risk and susceptibility geovisualization modeling framework.

In the United States, California ranks third in total area (U.S. Census Bureau, 2012), and has had the largest population of any state since the 1960’s (U.S. Census Bureau, 1996, 2011). There are 58 counties in California, and 8,057 census tracts (U.S. Census Bureau, 2019). WNV was first detected in California in 2003 (Reisen et al., 2004), and then received national attention for the high rates of the disease during the following two years (Jean et al., 2007). Results of WNV vector-borne environmental modeling in California (Kala et al., 2017) let to this study of combining socio-economic data with the results of the environmental model using multivariate geovisualization. This study utilized coarse-scale data (county level) of reported cases of WNV human incidence along with infected dead bird counts as the basis for estimating WNV risk. Fine scale environmental (120 m pixels) and coarse scale demographic data (census tract level) were used to define environmental and socio-economic factors for the study area. The study was conducted in two phases: (1) mosquito habitat modeling based on environmental factors and (2) geovisualization techniques based on socio-economic factors. Basemaps for this study were created either using (1) ArcGIS® software by Esri (ArcGIS® and ArcMap™ are the intellectual property of Esri and are used herein under license; copyright © Esri; all rights reserved; for more information about Esri® software, please visit http://www.esri.com), or (2) Topologically Integrated Geographic Encoding and Referencing system (TIGER) by the U.S. Census Bureau, which is in the Public Domain.

Study area and environmental and socio-economic factors affecting WNV

Reported human incidence rates for the study period by county were used to create a 3-dimensional database where the X and Y dimensions were the geographic centroids of each county, and WNV incidence rates for the county provided the Z dimension. Those data were then analyzed to generate a spatial 1-standard deviation ellipse (SDE), representing the contiguous region that contained 1-standard deviation of the reported human WNV incidence rates in California. SDE mapping is a common method used to identify spatial direction trends of attribute data associated with geographical features. It has been widely used for geographically identifying disease and crime trends (Chainey, Tompson & Uhlig, 2008; Wang, Shi & Miao, 2015; Leigh, Dunnett & Jackson, 2016; Al-Kindi et al., 2017; Ma et al., 2017; Polupan et al., 2017; Butkovic et al., 2019; Lu et al., 2019; Chen et al., 2020). We used SDE to identify the contiguous region that contained 1-standard deviation of WNV human incidence rates to focus on the counties in California that would most likely reveal previously unknown patterns of WNV risk and vulnerability, and defined that region as our study area.

Once the study area had been determined, socio-economic and environmental data were extracted from each census tract that intersected the ellipse. The dataset contained seven variables for each census tract. A single environmental variable (referred to in this study as “mosquito risk”) that represented the results of our earlier GWR model (Kala et al., 2017) was derived from analysis of environmental eight parameters (stream density, surface temperature, surface slope, cultivated land, developed land, road density, vegetation type, evapotranspiration rate). Mosquito risk was found to be statistically significantly related to annual WNV-infected dead birds sentinel data, averaged for the 2004–2010 (Kala et al., 2017). Annual WNV-infected dead birds sentinel data has been shown to be useful for estimating human WNV risk by multiple studies (Eidson et al., 2001a, 2001b, 2001c; Guptill et al., 2003; Mostashari et al., 2003; Ruiz et al., 2004; Johnson et al., 2006; Nielsen & Reisen, 2007; Patnaik, Juliusson & Vogt, 2007; Chaintoutis et al., 2014). The mosquito risk model resulted in a risk surface with a range of 0 to 10. Higher values indicate higher probability of WNV infected birds based on environmental conditions related to mosquito habitat. For the current study, mosquito risk was extracted for each of the census tract within the study area.

Numerous studies have shown that a susceptible population’s risk can be influenced by demographic and socio-economic conditions. For example, Ruiz et al. (2004) and Jean et al. (2007) suggest that the elderly are more susceptible because they have higher rates of weakened immune systems. Males and females may have differing vulnerabilities due to social history or lifestyle (Murray et al., 2006). Ruiz et al. (2004) also suggest that race/ethnicity or income influence vulnerability due to behaviors linked to lifestyle. For each census tract in the study area, the following data were extracted from 2010 Census data: percent of census tract’s population identified as male; percent of census tract’s population identified as white; percent of census tract’s population identified as black; percent of census tract’s population identified as Hispanic; median age of population in census tract, and; median household income in census tract.

Geovisualization techniques

This study utilized a spatially explicit exploratory approach for identifying the interaction between different environmental (mosquito habitat) and socio-economic (human demographic) processes occurring in each census tract within a 1-SDE. The approach consisted of utilizing the risk map with multivariate visualization techniques to facilitate the exploration and understanding of complex environmental and socio-economic patterns within the California data. The analysis was facilitated with SomVis, originally an open source Java application, that has now been ported to a web-based service (zillioninfo.com). SomVis was/is an integrated software tool consisting of three interactively linked visualizations that can help focus attention on patterns of similarity in complex data sets. The three visualizations used were: (1) a SOM (Kohonen, 2001) to perform multivariate analysis, dimensional reduction, and data reduction; (2) a PCP (Inselberg, 2002) to visualize the multivariate patterns with display; and (3) geographic mapping (GeoMap) to highlight clusters of specific interrelationships. The geovisualization tools of SOM and PCP have been adopted in many fields of science for exploring difficult high dimensional and non-linear problems as well as for visualization of multivariate problems (Edsall, 2003a, Koua & Kraak, 2004; Guo et al., 2005; Basara & Yuan, 2008; Kaur, Singh & Bahrdwaj, 2013; Brookes et al., 2014; Fanelli Kuczmarski et al., 2018; Mutheneni et al., 2018). These tools help to display the high-dimensional datasets, search for hidden relations among the complex set of variables and transform them into a 2-D pattern recognition problem.

Our study highlights the potential of combining these tools along with GIS to detect and analyze different hidden patterns within the complex multivariate data. The coupling of these techniques provides an interesting platform for analyzing larger datasets by integrating it into a spatially-explicit disease model or by using it for near-real time disease monitoring. This user interactive data exploration platform helps identify clusters of complex high dimensional datasets while preserving the topological relationships between data vectors.

I. SOM is used to reduce the dimensionality of data for data visualization purposes while retaining the most information contained within the database. It is a unique partitioning clustering method, which segments multivariate data into non-overlapping clusters and projects them on a two-dimensional layout. Koua & Kraak (2004) describe SOM as an unsupervised neural clustering technique that is useful in situations where the data volumes are large and interrelationships unclear. The approach involves partitioning the dataset where each element (in this case each census tract within the ellipse) is classified into one cluster out of a set number of desired clusters—49 in this study. Clusters contain elements that are similar to each other in terms of the observations for the statistically most relevant variables in the dataset. Some clusters may contain many elements (census tracts), while other may only contain a few, but census tracts within a cluster are more similar to each other than they are to census tracts in other clusters. Likewise, some clusters of census tracts can be more similar to other clusters, but are still different enough to be classified as different clusters according to the feature selection algorithm of SomVis. The clusters are then mapped onto a fixed grid of hexagons, in our case a 13-by-13 grid of hexagons to assist in data visualization. Each cluster is represented with a node (circle) whose diameter is linearly scaled according to the number of census tracts that it contains. Nodes are equally spaced in a two-dimensional space, and behind the nodes is a layer of hexagons, which are shaded to show the multivariate dissimilarity between neighboring nodes. Clusters falling on bright-tone hexagons are more similar to each other than those in darker tones of these hexagons.

II. A PCP maps n dimensional space onto a two-dimensional layout by using n equidistant parallel vertical axes, where n is the number of variables in the data set. Each vertical axis represents one variable and is linearly scaled using its minimum and maximum values. Each cluster is displayed as a horizontal polyline intersecting each of the vertical axes at the point that corresponds to the respective attribute value for this data element. The thickness of the polyline is proportional to the number of elements in the node (number of census tracts). The PCP can help visualize the data either using combinations of variables (cluster level) or for each individual variable (data item level).

III. Geographic mapping of which census tracts fall within any specific cluster or clusters provides a visual perspective of where the socio-economic and environmental variables of most interest are located. SomVis refers to these as a Geomap and they represent the spatial distribution of multivariate patterns. The Geomap provides a spatial perspective to clusters of similar variables identified using PCPs. These three visual components allow an array of user-controlled interactions that link spatial patterns to the underlying data.

Results

Our earlier study (Kala et al., 2017) found that the best-fitting mosquito habitat model that predicted number of WNV infected dead birds in all counties in California had an adjusted r2 of 0.71 (r2 = 0.75, p < 0.05). Those results agreed with other research (e.g., Beck et al., 1994) that found that understanding insect borne infectious disease risk is improved when considering spatial heterogeneity of the variables that affect the risk. Our current study, using the same mosquito habitat suitability modeling approach, also found that environmental modeling of environmental variables is improved when considering spatial heterogeneity of those variables. Figure 2 provides a WNV infection risk surface map based on the infected dead bird versus mosquito habitat model.

Figure 2 West Nile Virus (WNV) risk based on environmental context modeling (i.e., mosquito habitat risk).

Risk is represented by a unitless value that can theoretically range from a low of 0 (zero) to a high of 10 (ten), based on environmental variables that linked mosquito habitat to WNV infected dead birds as described in Kala et al. (2017).

In this study, we defined our study area as the 1-SDE of reported WNV incidence rates in California. California has 58 counties; 35 counties intersected the ellipse, representing a geographically contiguous area that represents approximately 67% of all WNV incidence rates. The counties within the ellipse averaged approximately 523,000 hectares in size. Defining this ellipse as our study area was a data reduction approach that allowed focusing on the most relevant WNV incidence rates. Figure 3 represents the counties, color coded by reported incidence rates along, with the 1-SDE based on incidence.

Figure 3 West Nile Virus human incidence rate by county with a 1-standard deviation ellipse superimposed.

California has 58 counties; 31 counties are contained within or intersect with the 1-standard deviation ellipse. Colors represent quintiles of reported human incidence of WNV. Built using ESRI ArcGIS® and ArcMap™ basemap files (ESRI, Redlands, CA, USA). Sources for basemap: National Geographic, Esri, Garmin, HERE, UNEP-WCMC, USGS, NASA, ESA, METI, NRCAN, GEBCO, NOAA, increment P Corp.

Socio-economic (demographic) variables were extracted for all census tracts within the ellipse. California has 8,040 census tracts, with 1,133 intersecting the 1-SDE. The census tracts within the ellipse averaged 8,780 hectares in size. Environmental and socio-economic data were considered simultaneously with SOM analyses. The resultant SOM identified 49 distinct nodes of census tracts (Fig. 4).

Figure 4 Census tracts (1,133) within the 1-standard deviation ellipse of human West Nile Virus incidence rate.

Built using ESRI ArcGIS® and ArcMap™ basemap files (ESRI, Redlands, CA, USA) and Topologically Integrated Geographic Encoding and Referencing system by U.S. Census Bureau. Sources for basemap: National Geographic, Esri, Garmin, HERE, UNEP-WCMC, USGS, NASA, ESA, METI, NRCAN, GEBCO, NOAA, increment P Corp, U.S. Census Bureau.

Each SOM node shown in Fig. 5 (indicated with colored circles) represents a cluster of census tracts that are most similar in terms of all seven variables. The diameter of each node represents the number of census tracts in the node. To illustrate how geovisualization can be used by public health planners, two specific nodes are highlighted for discussion. First, the cluster that contains census tracts with the highest median age is highlighted (labeled as cluster 1 and green in color), and is of interest because it is a variable that has been described as representative of the most vulnerable population (the elderly) to WNV health issues (e.g., Campbell et al., 2002). Second, the cluster that contains census tracts with the highest environmental WNV risk (mosquito habitat) based on the GWR model is highlighted (labeled as cluster 2 and blue in color) because of the statistically significant relationship to WNV infected dead bird count.

Figure 5 Self organizing map representing 49 nodes with valid combination of contextual and compositional parameters from 1,133 census tracts.

Size of node (circle) reflects how many census tracts in the cluster. Darker gray shading of background hexagons represents more dissimilarity to nearby clusters.

Once SOM nodes are defined, a PCP can be developed to explore the interaction between different environmental and socio-economic risk factors. The PCP shows seven vertical axes representing each of the variables under consideration, and 49 polylines representing clusters of census tracts that are most similar to each other for those seven parameters. Figure 6 represents the PCP with the polyline for cluster 1 (census tracts with the highest average median age) highlighted in green. The PCP indicates that the census tracts contained within this cluster average: (1) the lowest percent male (~45%); (2) a moderate household income (~$60,000); (3) the lowest percent Hispanic (~9%); (4) the highest median age (~51 years); (5) nearly the highest percent white (~89%); (6) a low percent black (~1%), and; (7) a moderately high mosquito habitat risk (~6.5).

Figure 6 Parallel coordinate plot showing 49 polylines representing each cluster; green highlighted polyline represents cluster with the highest median age.

Compositional parameters include average values of all census tracts in cluster for: percent of population that is male, median household income, percent Hispanic, median age, percent white, percent black. Contextual parameters include mosquito habitat risk based on environmental parameters related to West Nile Virus infected dead birds. Bold numbers on each axis represent the maximum average value and the minimum average value for the 49 clusters.

Turning to the cluster with census tracts that average the highest environmental risk (mosquito habitat suitability),) this cluster can be visualized with the polyline shown in blue in Fig. 7. The PCP indicates that the census tracts contained within this cluster average: (1) a moderate percent male (~49%); (2) a moderately low household income (~$55,000); (3) a moderately low percent Hispanic (~19%); (4) a moderate median age (~36 years); (5) a moderately high percent white (~79%); (6) a moderate percent black (~4%), and; (7) the highest mosquito habitat risk (~7.1).

Figure 7 Parallel coordinate plot showing 49 polylines representing each cluster; blue highlighted polyline represents cluster with the highest mosquito habitat risk.

Discussion

After finding a significant relationship between environmental variables related to Culex mosquito habitat and the number of dead birds infected with WNV, we examined human incidence rates in California to extract socio-economic data (population demographics related to WNV susceptibility). Our goal was to use geovisualization techniques to explore the combination of both environmental and socio-economic information to better understand this vector borne infectious disease. Out of the very large number of questions that could be explored with geovisualization, we highlighted two specific ones here: (1) what are the characteristics of the California cluster that represents the census tracts with the highest median age, and; (2) what are the characteristics of the California cluster that represents the census tracts with the highest mosquito habitat risk. Many other questions can be explored once the data are extracted, but to illustrate the technique, we will focus on these two questions.

For example, the cluster that contains the census tracts with the highest median age (~58 years) can be visualized in the SOM—it is the node highlighted in green and labeled “cluster-1” in Fig. 5. In the SOM, this node is represented with a circle of moderate diameter indicating that it contains a moderate number of census tracts compared to other nodes, and it is located in a moderately toned gray area indicating that it is moderately dissimilar in multivariate space to other nodes in the study area. “cluster-2”, highlighted in blue in Fig. 5, represents the node that contains the census tracts having the highest mosquito habitat risk. The node’s diameter is relatively large, indicating that it contains a large number of census tracts compared to other nodes. Like cluster-1, cluster-2 is located in a moderately toned gray area, indicating moderate dissimilarity to other nodes.

The 49 clusters were then analyzed with PCP, allowing visual inspection of the characteristics of the input parameters of each cluster. Cluster-1 (composition includes highest median age) is highlighted as a green polyline in Fig. 6. Following the polyline for cluster-1 indicates that in addition to the highest median age, it also contains a group of census tracts with: the lowest percent males; a moderate median household income; the lowest percent Hispanic; nearly the highest percent white; nearly the lowest percent black; and a moderately high mosquito habitat risk. This visualization may suggest to public health planners that overall this cluster may not be as vulnerable to WNV as the initial reaction for concern for census tracts with the highest median age might imply.

Cluster-2 (environmental context shows highest WNV mosquito habitat risk) is highlighted as a blue polyline in Fig. 6. While this group of census tracts represent the highest WNV mosquito habitat risk, they contain relatively moderate levels of the six population socio-economic parameters. Implications of the information from this cluster may also be important to inform public health planning.

Clusters can also be viewed spatially for additional geographic insight. Figure 8 provides a map of census tracts with the two highlighted clusters isolated. Census tracts colored green (n = 19) represent those with the highest median age. The non-contiguous nature of the census tracts associated with this cluster indicates that they are only similar based on their non-spatial attribute characteristics rather than because of geographical location or autocorrelation. In contrast, the cluster that contains census tracts with the highest WNV mosquito habitat risk (colored blue, n = 30) tend to be concentrated in geographic space. This spatial insight would be valuable to public health planners who may be planning interventions.

Figure 8 Geomap showing spatial context of census tracts contained in the cluster (#1 in the self-organizing map (SOM)) with the highest median age (green) and the census tracts in the cluster (#2 in the SOM) with the highest mosquito habitat risk (blue).

Built using Topologically Integrated Geographic Encoding and Referencing system basemap files. Sources for basemap: U.S. Census Bureau.

The results from this exploratory analysis suggest that further investigation is required to fully understand the relationship between age and WNV risk. As mentioned above, studies have suggested that elderly people are more vulnerable to WNV, but others such as Carson et al. (2012) shows that WNV infection was greatest for the younger population. It would be simple for public health planners to want to visualize the cluster that contains the census tracts with this composition (lowest median age), and use the PCP to visualize the characteristics of that cluster. If, on the other hand, the planner would rather focus on WNV human incidence rates, census tracts that occur in areas with the highest incidence rates might drive the visualization. For example, Glenn County (near the northern edge of the SDE in Fig. 3) reported the highest WNV rate during the study period, so the planner might be interested in finding the cluster(s) that contain the census tracts of this county. This county has six census tracts, and Fig. 9 shows the PCP highlighting the five clusters that contain those census tracts. Two of the six census tracts fall within a single cluster (highlighted in pink), but the other four census tracts each fall in four separate clusters. These polylines, representing all five clusters that occur in the county with the highest WNV incidence rates, reveal an unexpected pattern. These five clusters, all representing distinct combinations of environmental and socio-economic data, all have a relatively low WNV mosquito habitat risk. This newly revealed pattern reinforces a suggestion that WNV disease, like other vector-borne infectious diseases, may not necessarily be contracted in the location where a person lives, but rather where they may have traveled to locations that represent higher risk areas. The pathogen may be contracted during outdoor activities in a higher risk area, and then later their disease is diagnosed by the victim’s local physician and reported using a local address. While that idea is a common-sense caveat in many vector-borne research conclusions (see for example: Atkinson et al., 2012, 2014; M’ikanatha & Iskander, 2014; Riddle, 2020), this data mining geovisualization analysis provides some initial evidence to that effect. The low correspondence between WNV habitat risk (Fig. 2) and actual incidence of WNV disease in the population (Fig. 9) highlights why a geographically based visualization of the relationships between environmental and socio-economic data may be useful.

Figure 9 Parallel coordinate plot highlighting the five clusters found in Glenn County, the county with the highest human incidence rate of West Nile Virus.

Pink line represents the only cluster that contains more than one census tract in Glenn County.

Additionally, the public health planner may want to explore all clusters represented in Glenn County in order to understand census tracts outside of Glenn County. For example, if the focus is on the only cluster in Glenn County that contains more than one census tract, the planner may want to explore other census tracts outside of Glenn County that are contained in that specific Glenn County cluster. That cluster represents 21 census tracts in the study area (see Fig. 10), but they don’t have any spatial relationship to each other. After visualizing this pattern, public health practitioners may plan on providing heightened information on detecting WNV symptoms to physicians in those census tracts, since the environmental and socio-economic patterns uncovered in those census tracts are highly related to those in Glenn County, where WNV incidence was the highest.

Figure 10 Census tracts, highlighted in pink, within 1-standard deviation ellipse that are in the same cluster that contains more than one census tract found in Glenn County.

Built using Topologically Integrated Geographic Encoding and Referencing system basemap files. Sources for basemap: U.S. Census Bureau.

These examples of geovisualization data mining to explore environmental and socio-economic data related to WNV disease in California represent only a few of the many questions that public health planners may pose. The planners most familiar with the spatial, temporal and historical setting of WNV in California will almost certainly generate different questions. Other infectious diseases in other areas will also generate specific questions to be explored by public health practitioners. Geovisualization will likely provide unique insights.

Conclusions

Developing new analytical models that combine environmental and socio-economic model for infectious disease planning is difficult because the data are often collected at differing scales, using differing boundaries, and under differing research contexts, each of which might help explain pieces of an infectious disease independently, but in aggregate may provide much better insight. This article suggests that an exploratory geovisualization process can help planners understand the interplay between environmental and socio-economic data prior to embarking on the difficult development of an analytical model that accounts for these disparities.

This study explored the use of geovisualization techniques to uncover patterns in large, complex data sets that would be difficult to otherwise discover. WNV was used as a case study to explore this question—California became the center of United States attention in 2004 and 2005 due to high rate of disease incidence. Geovisualization allowed combining the spatially explicit environmental factors (mosquito habitat risk) with socio-economic data (population demographics) in a data mining context to find previously unknown data clusters at the census tract level. Major challenges for multivariate geospatial mapping include large data volumes, high dimensionality, and the perception of complex patterns (Guo, 2009). The research reported here utilizes a spatially explicit exploratory approach that combines geovisualization, spatial analysis, and computational methods for identifying the interaction between different environmental and socio-economic factors. There are multi-level dynamics involved in a disease transmission including complex environmental procedures and the population dynamics. Our research has explored the use of spatially explicit geovisualization techniques for identification of interesting clusters (based on their multivariate similarity) for future investigation. Our results suggest that the visualization of similarity clustering of multivariate attributes facilitates the analysis of complex data. It also helps expose the underlying spatial processes that may result in differential risks. Another advantage of this approach is that patterns found in voluminous and complex epidemiological data can provide more focused opportunities for analysis and interpretation by experts in that field. With an interactive user platform, geovisualization techniques can efficiently obtain new knowledge from the data and become an important hypothesis-generating tool in public health research. Understanding underlying environmental and socio-economic characteristics for the occurrence of WNV, or any infectious disease, is important for mitigating future outbreaks.

We have shown a few examples of how geovisualization could be used by public health planners to better understand and respond to an infectious disease outbreak. This approach found 1,133 census tracts within our study area of WNV incidence in California, and classified those census tracts into 49 clusters where each cluster contained census tracts that were more similar to each other in terms of WNV environmental and socio-economic parameters, than to the census tracts represented in all other clusters. Examples of several interesting patterns were revealed. For example, the cluster that had census tracts with the highest average mosquito habitat risk only had mid-level median age levels. Had there been a cluster that had both the highest mosquito habitat risk and the highest median age, public health planners might choose more intense intervention measures in those census tracts. Another interesting pattern uncovered was that census tracts in the county that had the highest reported incidence of WNV had relatively low mosquito habitat risk. This might lead to a speculation that demographic and socio-economic parameters should be weighted more importantly than mosquito habitat risk when developing public health plans. Likewise, this pattern might suggest other factors like poor links between modeled mosquito habitat risk and WNV risk in areas outside the training set data or spatial biases in recording effort operating differently at the county level and the census tract level could be at play. Focusing on those ideas through geovisualization may reveal other unknown patterns.

This article represents a case study that utilized a retrospective view of a WNV outbreak in California in the mid 2000’s. At that time, geovisualization tools were quite limited and not often used by public health practitioners. Now that the tools are more available, and much easier to use, a future research program that explores using geovisualization in near-real time during an outbreak is appropriate. Infectious disease outbreaks occur frequently, and rapid planning and response are always desirable. Many of these outbreaks are not well understood, and adequate interventions could certainly benefit from data mining, geovisualization approaches. For example, at the time of this writing the Coronavirus (COVID-19) was first reported to the public on 31 December 2019, after the outbreak was first detected in Wuhan City, China (CDC, 2020). By mid-February 2020, tens of thousands of cases were reported and news of the virus spreading outside of China started appearing in January 2020. This outbreak will clearly create a large and complex dataset, and public health planners would certainly benefit if they were able to explore geospatial patterns in that dataset in near-real time.

Supplemental Information

Supplemental Information 1 WNV risk and susceptibility in central California geovisualization modeling project dataset.

This dataset was developed to support research intended to develop a spatially explicit model that explores environmental data related to the risk of exposure to WNV, and the susceptibility to WNV disease based on demographic data of the potentially affected population. The model was developed and then tested on census tracts in an identified 1-standard deviation of WNV incidence in central California. The dataset contains (1) U.S. Census Bureau demographic data for 1,133 census tracks in the ellipse, and (2) and average mosquito habitat risk data for each of those census tracks based upon the model described by West Nile Virus risk based on mosquito habitat model as described in: Kala AK, Tiwari C, Mikler AR and Atkinson SF, 2017, A comparison of least squares regression and geographically weighted regression modeling of West Nile Virus risk based on environmental parameters, PeerJ 5:e3070; DOI 10.7717/peerj.3070

Click here for additional data file.

Supplemental Information 2 Short video showing example of interactive geovisualization.

A visual explanation of the plethora of geovisualizations that can uncover patterns in complex data sets.

Click here for additional data file.

The authors would like to acknowledge the pain and suffering of victims of West Nile Virus, as well as all other vector-borne infectious diseases.

Additional Information and Declarations

Competing Interests

Author Contributions

Data Availability

The authors declare that they have no competing interests.

Abhishek K. Kala conceived and designed the experiments, performed the experiments, analyzed the data, prepared figures and/or tables, authored or reviewed drafts of the paper, and approved the final draft.

Samuel F. Atkinson conceived and designed the experiments, performed the experiments, analyzed the data, prepared figures and/or tables, authored or reviewed drafts of the paper, and approved the final draft.

Chetan Tiwari conceived and designed the experiments, performed the experiments, analyzed the data, authored or reviewed drafts of the paper, and approved the final draft.

The following information was supplied regarding data availability:

The data is available in the Supplemental File.

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
