# Peer review of "Exploring the socio-economic and environmental components of infectious diseases using multivariate geovisualization: West Nile Virus"

_PeerJ, doi:10.7717/peerj.9577_

## Round 0.1 · original submission · Major Revisions

Three reviewers returned their comments. I am much in line with the third reviewer's concerns of the research question and justification of the methodology. For example, you shall provide literature evidence of why the 1st standard deviation ellipse can help extract the census tracts for the GWR analysis, and why and how SOM and PCP can help improve our understanding of the disease distribution associated with population vulnerability. Therefore, I invite the authors to revise the manuscript to address these concerns and other ones. A major restructure of the manuscript is thus expected. A rebuttal letter should be submitted with the revised manuscript.

Reviewer 1 ·

Basic reporting

Manuscript is well written with introduction and background.
Figures are relevant, good resolution and properly labeled.

References are sufficient and relatively recent.

Structure, figures and tables are per the journal’s specification. Raw dataset is also provided.

Experimental design

Research question is well defined, relevant and meaningful. In order to improve the predictability of the model, authors accounted for spatial autocorrelation and non-stationarity using GWR.

While the SOM is explained, its algorithm is not provided. Authors simply introduce the mechanism of reducing the dimensionality of data, without providing the detail of the cluster process. Therefore, the cluster algorithm remains unknown.

Validity of the findings

Authors indicated in the introduction that they included demographic data in the GWR model to improve the model fitting. But I am not sure how the authors can conclude that the demographic data are enhancing the model performance without demonstrating the process. Also, only two clusters are analyzed which may not be adequate to represent the pattern of WNV infection.

The conclusions could have been stronger if the results were validated.

Reviewer 2 ·

Basic reporting

• Writing is generally good
• Line 58 should be Diaz, Roux, and Mair or Diaz et al.?
• In general need to be consistent in citation style
• Line 68 ‘spread’ could be ‘primarily spread’
• Are there missing references at the end? E.g. I don’t see Wimberly
• Don’t see Ozdernal, Kilpatrick in list of refernces but in Table 1
• Need to check all citations are referenced
• Line 273 take out ‘the’ in ‘the a moderately’
• Generally I don’t think it is the best to call demographic variables ‘non-spatial’
• Maps aren’t very clear or informative (i.e. Figures 8 and 10). Purple background no use. How about having a base map for context? Reduce visibility (thickness) of census tract outlines

Experimental design

• Lines 73-77, I don’t believe there has been sufficient establishment to make this statement. Could there be something more to support this statement? What is meant by non-spatial factor? Wouldn’t it pay to establish here that technique wise there has been a lack of use of geovisualization? Actually I see more of this below.
• Not enough justification for environmenta/climatic variables in relation to relevant vector species in California.
• Unclear on exactly the way the risk surface (Figure 2, Table 1) was calculated. Lines 133-142 could be expanded upon.

Validity of the findings

• The findings are somewhat limited – e.g. in the Discussion section stretching a bit, strengthen by tying a bit more to literature
• Also discuss the fact that the work would be more valuable if there were WNV human incidence data at a spatially more specific level (e.g. census tract – e.g. DeGroote et al.)
• The WVV risk model (Figure 2) seemingly doesn’t have a high correspondence with the actual occurrence of WNV (Figure 3)

·

Basic reporting

The introduction is nicely written and I have some minor comments below. However, as described in "Experimental design" and "Validity of the findings", I have major reservations about the analytical methods used and their interpretation.

Introduction
Line 76. Please can the authors clarify the distinction between “contextual and compositional factors”, possibly since both mosquito habitat availability and age structure of the population at risk could be considered to be contextual or compositional factors. For example, there may be varying composition of mosquito vector species across the landscape within the community or the landscape composition may vary in the amount of larval mosquito habitat available. Would social versus environmental factors be better?
Line 90. “number WNV-positive dead birds”, insert “of” after “number”
Line 91 to 93. Can the authors say something more substantial about the results of their prior study here e.g. about the role of different environmental factors in explaining WNV risk?
Line 97. Briefly describe here how the composition of the susceptible human population may be informative about WNV risk, referring to prior papers, to give the reader context.

Results, Line 221. The authors seem to suggest that they are among the first to find that including geographical environmental factors improves prediction or understanding of vector-borne disease patterns, referring to Beck’s 1994 finding. There are some 10s to 100s of papers that have shown this finding since which could be cited in the discussion

Experimental design

1. The research question is not well defined.

The study initially postulates that both the underlying environmental context and the susceptible population’s composition should be understood to adequately predict a vector borne disease infection risks, a point which is relatively novel and with which I agree. However, the specific question in relation to the WNV study system is not well defined and the analyses selected do not demonstrate the linkage between these factors well at all. The outputs have limited value for public health planning, contrary to the authors suggestions.
Line 77 states “we hypothesize that geovisualization techniques to explore these relationships will result in a more complete understanding of the complex patterns related to this disease.”
Since the authors do not explicitly relate social factors and mosquito habitat to WNV incidence within a spatial model, and assess the variance explained by these two sets of factors they have not yielded new understanding of risk factors. It is not clear to me why the authors have chosen to use tools like self-organizing maps and parallel coordinate plots (line 98), rather than simply adding the human population factors to the prior geographically weighted regression.

2. Analytical methods are not appropriate or rigorous.
Line 180. The authors do not adequately describe why they want to create a self-organizing map, saying that it is to “reduce the dimensionality of data for data visualization purposes”. This data reduction seems unnecessary and unlikely to improve interpretation, since they authors only have 7 variables per census tract and >8000 census tracts in their sample, thus the number of data points per candidate predictor are large and should allow the predictor effects to be robustly estimated in a GWR model or other spatial mixed modelling approach.

2.1. The self-organizing map results (Line 268 to 294 and Fig. 5) are very hard to interpret and of low value for public health because these are purely statistical clusters, where neither the cluster membership, nor the social or mosquito habitat variables that define clusters, have been related to WNV incidence for the tracts in this setting. Very many clusters are created, some of which are very small, covering very little of the area of interest. Only 19 and 30 census tracts out of over 1000 tracts fall into the two clusters highlighted for their public health importance. What practical advice could be given to public health decision makers on the basis of highlighting two clusters on a map with interacting mosquito and social factors, the latter of which have unknown predictive power for WNV incidence? From the % ethnicity variables in the PCP plots, it seems that non-orthgonal variables are being used in the clustering which further reduces the interpretability of the clusters. In terms of the vulnerability of different age groups to severe clinical outcomes of WNV, I would also question whether the median age is the most appropriate metric of age structure to use. Isn’t it more important for health planning to know how the absolute number of individuals from vulnerable age groups varies between census tracts with different mosquito habitat risk?


2.2 Line 157. The way that the high incidence data area is created for the SOMs, from the county level human incidence data, is extremely strange. The incidence data are treated as reflective of the centroids only of a county rather than of the county as a whole, and used to create an elliptical high incidence area. The authors do not make it clear why they wanted to reduce the census tracts to only those in high incidence areas (Line 231) in the first place but this method of finding the high incidence area is not as defensible as simply using the admin boundaries of the counties from which we know the data are recorded. Overall, when studying environmental and social risk factors for vector-borne diseases in a spatial framework, you may be missing important variation if you reduce your study area to only high incidence areas.

2.3 Line 139. The methods and results for the mosquito habitat model are very unclear. The landscape variables in Table 1 are not described in enough detail for someone else to replicate the study. E.g. Is “temperature” the annual mean air temperature per tract? What resolution and which data source was this derived from? Were the stream amounts lengths of stream per census tract? Were the cultivated and developed lands the proportional or absolute amount of such land within a particular buffer zone of the census tract. How were the weights for the overlay selected and validated against mosquito or WNV incidence data, using which statistics? Line 225. The authors should give some statistics to demonstrate the variance explained in observed census-tract level WNV infection risk by their mosquito habitat model, similar to those they provide from their prior model, rather than just providing the map in Figure 2. Without this information, it is impossible to assess the value of the map for public health planning.

3. Minor points requiring clarity in the methods are how large are census tracts relative to counties on average and what were the time periods from which the different types of WNV incidence data were drawn.

Validity of the findings

4. Because of the methodological short-comings above, especially the lack of validation statistics for the mosquito model, the small geographical size of the clusters under discussion, and the unproven predictive ability of the variables defining the clusters for incidence, most areas of the discussion and conclusions are not well founded. I give three specific examples here.

4.1 Line 338 “This newly revealed pattern reinforces a suggestion that WNV disease, like other vector-borne infectious diseases, may not necessarily be contracted in the location where a person lives, but rather where they may have traveled to locations that represent higher risk areas.”
This finding has been widespread across vector-borne diseases and has led authors to map human activities that lead to exposure. A much wider range of literature could be cited here.

4.2. Line 400 “Another interesting pattern uncovered was that census tracts in the county that had the highest reported incidence of WNV had relatively low mosquito habitat risk. This might lead to a speculation that demographic compositional parameters should be weighted more importantly than mosquito habitat risk when developing public health plans.”

This could easily be explained by other factors like poor links between modelled mosquito habitat risk and WNV risk in areas outside the training set data or spatial biases in recording effort operating differently at the county level and the census tract level.

4.3. Line 382. Our results suggest that the visualization of similarity clustering of multivariate attributes facilitates the analysis of complex data.
I can't agree with this statement. Due to the low interpretability of the analyses used and their low value for public health interventions, I don't agree that there is benefit to the literature from this study in its current form.

Additional comments

I was excited by the introduction to your paper since I agree that frameworks for vector-borne diseases badly need to link environmental hazard with interacting social factors that determine exposure and vulnerability. However, as I have described above, I don't think that the analytical methods you have chosen are appropriate or robust for such linkage. I would very much hope that the comments above help in some way in the re-framing of your models so that you can re-formulate the paper and outputs for public health.

---

## Round 0.2 · Minor Revisions

You have addressed all the reviewers' previous concerns in the revision, and the quality of the manuscript has been greatly improved. I suggest one minor item to be fixed for all the maps in the figures: scale bars are needed.

---

## Round 0.3 · accepted · Accept

Your manuscript has met the quality for publication in PeerJ - Life & Environment.